# Modulus adaptive lubricating prototype inspired by instant muscle hardening mechanism of catfish skin

Yunlei Zhang[1,2], Weiyi Zhao[1,2], Shuanhong Ma 📀 [1✉], Hui Liu[1,2], Xingwei Wang[1], Xiaoduo Zhao[1], Bo Yu[1], Meirong Cai[1] & Feng Zhou[1✉]

In nature, living organisms evolve unique functional components with mechanically adaptive compatibility to cater dynamic change of interface friction/lubrication. This mechanism can be used for developing intelligent artificial lubrication-regulation systems. Inspired by the muscle hardening-triggered lubrication of *longsnout catfish*, here we report a modulus adaptive lubricating hydrogel prototype consisting of top mucus-like hydrophilic lubricating layer and muscle-like bottom hydrogel that can stiffen via thermal-triggered phase separation. It exhibits instant switch from soft/high frictional state (~0.3 MPa, $\mu$~0.37) to stiff/lubricating state (~120 MPa, $\mu$~0.027) in water upon heating up. Such switchable lubrication is effective for wide range of normal loads and attributed to the modulus-dominated adaptive contact mechanism. As a proof-of-concept, switchable lubricating hydrogel bullets and patches are engineered for realizing controllable interface movements. These important results demonstrate potential applications in the fields of intelligent motion devices and soft robots.

[1] State Key Laboratory of Solid Lubrication, Lanzhou Institute of Chemical Physics, Chinese Academy of Sciences, 730000 Lanzhou, China. [2] College of Materials Science and Opto-Electronic Technology, University of Chinese Academy of Sciences, 100049 Beijing, China. ✉email: mashuanhong@licp.cas.cn; zhouf@licp.cas.cn

In nature, organisms have evolved functional organs with mechanically adaptive compatibility to accommodate complicated living environment, which provides inspiration for developing artificially intelligent materials systems[1–3]. The ingenious strategy for coupling mechanically adaptive contact state and surface/interface interactions of organs, plays an important role for realizing excellent friction/lubrication performance in the dynamic contact process during locomotion[4–6]. For example, the excellent lubrication performance of natural articular system under dynamic harsh loading/unloading condition is obtained by the mechanically adaptive cartilage layer and highly hydrated surface[7–9]. Meanwhile, considering the intelligent design concept, artificially responsive material systems such as supramolecules[10–12], hydrogels[9,13–16], and polymer brushes[17–20] have been designed to regulate the surface interactions for achieving desirable friction force. However, these systems commonly rely on the change of surface molecular states or hydration degree of responsive layers, while leaves the friction contribution from tunable contact states out of consideration. Furthermore, the poor mechanical tolerance of these responsive layers commonly results in a limited regulation magnitude of friction force. In fact, mechanical properties around materials surface also affect the interface contact state and thus control the friction force[21]. Nevertheless, inherent mechanical properties of materials is difficult to be changed directly/sharply once prepared, limiting the adaptive manipulation of contact states. So, it is still a great challenge to construct intelligent friction/lubrication materials system, based on synergistic regulation strategy coupling mechanical adaptive contact states with surface interactions.

Intriguingly, one synergy inspiration can be found in nature. Fish evolve unique locomotion and lubrication mechanisms based on the highly hydrated surface mucus and enhanced by the modulus adaptive muscles. Mucus secreted on *longsnout catfish* skin is one kind of highly hydrated biomacromolecules and provides them with slippery surface to get rid of predators, reduce swimming drag and adapt to the aqueous living environment (Fig. 1a)[22–24]. So, it is difficult to catch fish with mucus (Fig. 1b-i). However, an undeniable fact is that *longsnout catfish* can still be held when they are calm (Fig. 1b-ii), while it becomes too slip to be caught when struggling (Fig. 1b-iii). Actually, the different capture phenomena are related to the evolution of interface coefficient of friction (COF) between fish skin and hands, resulting from the adaptive switch of skin's muscle modulus (Fig. 1b-iv)[25,26]. When perceiving external stimuli, fish skin achieved an adaptive modulus switch along with the contraction and stratching of the red and white muscles inside body[5,27–29]. In addition, due to the lack of hard fish scales acting as a mechanical barrier for the transmission of these internal modulus changes to external skin in *longsnout catfish*, the stiffening of the muscles leads to an instant modulus increase in skin system[30,31]. As a result, although the surface mucus can reduce the COF, the adaptive and switchable enhancement of skin modulus can decrease the contact area and the mechanical deformation to further minimize the surface friction force[32–34], which is beneficial for fish to escape from dangerous situations or adapt to its external surroundings.

Inspired by the switchable lubrication of *longsnout catfish*, an interface friction force control strategy based on the modulus switch was proposed. Herein we reported one kind of layered thermal-triggered intelligent lubricating material, modulus adaptive lubricating hydrogel (MALH) (Fig. 1c), consisting of the top hydrophilic lubricating layer (mimicking the slippery surface mucus of fish) and bottom thermal-stiffen phase-separation hydrogel layer (mimicking the hardening effect of fish muscles). The bottom phase-separation layer can realize magnitude change (~400 times) of elastic modulus from soft state (~0.3 MPa, 20 °C)

to rigid state (~120 MPa, 80 °C), similar to the muscle before and after contracting or tensing[35,36], which enabled the material to achieve switchable lubrication behavior (COF, $\mu$, from ~0.37 to ~0.027) in water upon heating. Correspondingly, thermal-triggered switchable lubricating bullet was engineered to demonstrate its potential application. Finally, an intelligent lubrication patch, named as modulus adaptive switchable lubrication device (MASLD), can be successfully fabricated by MALH. MASLD can be assembled on mobile devices to control their movement under wet environment. These results showed the strategy of coupling adaptive modulus switch with interface lubrication can enhance the interface friction-control, and implied a promising applicability of MALH in soft robots and intelligent lubrication systems.

## Results

**Concept and construction of MALH.** Figure 2a showed the preparation process for MALH. Firstly, the P(AAc-co-HEMA-Br) hydrogel was synthesized by UV polymerization of the precursor solution containing acrylic acid (AAc), 2-hydroxyethyl methacrylate-bromine (HEMA-Br, Supplementary Fig. 1), crosslinker (N,N'-Methylenebisacrylamide, MBAA), initiator (α-Ketoglutaric acid) and DI water, and then followed by post-treatment in calcium acetate (CaAc) solution to form a temperature-responsive phase-separation PAAc-CaAc network[37]. Subsequently, the P(AAc-CaAc-co-HEMA-Br) phase-separation hydrogel was immersed into reaction solution containing monomer (3-sulfopropyl methacrylate potassium (SPMA)) to perform atom transfer radical polymerization (ATRP) for generating surface lubricating layer. As a result, the P(AAc-CaAc-co-HEMA-Br-PSPMA) hydrogel (MALH) was successfully prepared. The Attenuated Total Reflection Flourier transformed Infrared Spectroscopy (ATR-FTIR) was used to confirm the components of MALH (Supplementary Fig. 2a). The peak appeared at ~1050 cm$^{-1}$ was the stretching vibration signal of sulfur-oxygen double bond (-S=O), indicating the successful grafting of PSPMA polyelectrolyte brush on the surface of hydrogel.

The mechanism for modulus change of MALH resulted from the strategy of enhancing the electrostatic interaction in hydrophobic media at high temperature (Fig. 2a). In details, the hydrophobic residues (acetate) in MALH network dehydrated at high temperature to form hydrophobic regions[38–40], for which they can remarkably reduce the local permittivity of the media, induce phase separation and strengthen the electrostatic interaction of the nearby charged residues (Ca$^{2+}$-COO$^-$)[37,41]. As a result, MALH underwent a rapid phase transition from soft gel to rigid plastics. Further analysis from ATR-FTIR showed that the water-hydroxyl binding peak signal at ~3300 cm$^{-1}$ decreased sharply for P(AAc-CaAc-co-HEMA-Br) hydrogel (or the substrate of MALH) after phase separation (Supplementary Fig. 2b), due to strong dehydration effect from hydrophobic residues at high temperature. By contrast, no obvious decrease of this peak signal was detected on MALH surface, indicating the good hydration ability of surface lubricating layer. Correspondingly, as shown in Fig. 2b, the bulk thermal-stiffen layer of MALH served as a stimulus-responsive matrix for mimicking the fish's muscle-stiffen mechanism, while the highly hydrated surface lubricating layer served as the mucus-like slippery surface. Therefore, similar to a struggling fish, MALH can manipulate the interface contact state and COF through the phase-separation-induced modulus change under thermal stimulus, so as to achieve switchable lubrication like fish.

**Switchable mechanical property of MALH.** The phase separation of MALH lead to two completely distinct mechanical states

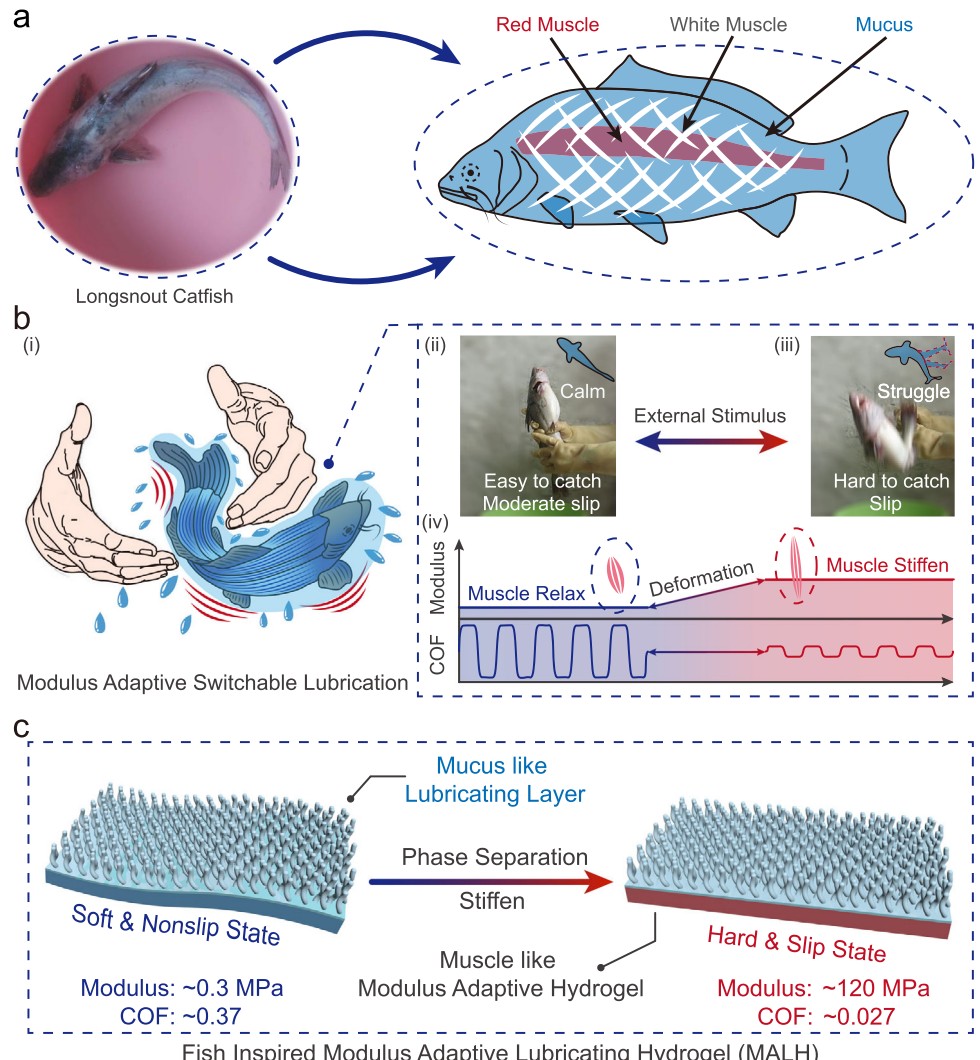

**Fig. 1 Concept for designing MALH. a** The photography of a *longsnout catfish* in water and the corresponding schematic diagram of its body contour, mucus and muscle. The schematic diagram was reproduced with permission from ref. [29]. Copyright 1988 Springer Nature. **b** Modulus adaptive switchable lubrication when fish being caught: (i) schematic illustration of escaping process of fish being caught, (ii) fish was calm and easy to catch, (iii) fish was struggling and hard to catch, and (iv) the correlationship between the skin muscle modulus and COF to demonstrate the modulus adaptive switchable lubrication phenomenon of fish. **c** Schematic diagram showing the conceptual structure of MALH inspired by instant muscle-hardening mechanism of fish.

in wet environment: soft state (low temperature) and rigid state (high temperature) (Fig. 3a). MALH can switch instantly from soft/transparent (low modulus) state ($T = 20\,°C$) to rigid/opaque (high modulus) state ($T = 80\,°C$) within 1 s (Fig. 3ai-iv and Supplementary Movie 1), due to the cooperative effects of hydrophobic interaction and ionic interaction. This transition was similar to the modulus change of the muscle before and after contracting or stretching (Fig. 3aiii)[35,36,42]. In addition, MALH can be fixed into specific shapes in rigid state to bear heavy loads (200 g weight) without deformation (Fig. 3av-vi), implying its good load-bearing capacity. Figure 3b showed temperature-dependent strain-stress curves of MALH from 20 to 80 °C, indicating its tunable mechanical properties. At low temperature, MALH was soft with a large strain and a small breaking strength, whereas at high temperature, it was rigid with a small strain and a high breaking strength. It was found that the content change of HEMA-Br (ATRP initiator) showed little effect on modulus and tensile stress of MALH both in soft state and rigid state (Fig. 3c, d). However, high content of HEMA-Br (5 wt%) in monomer solution would cause slight aggregations (Supplementary Fig. 3), which

were not beneficial for sample preparation, so 3 wt% of HEMA-Br was chosen.

The modulus of MALH before and after phase separation were ~0.3 MPa (soft state) and ~120 MPa (rigid state), respectively. The stiffness change magnitude of MALH was ~400 times. Correspondingly, the modulus regulation range of MALH system can match the mechanical properties of diverse biological tissues, compared with other reported modulus-switching systems (Supplementary Fig. 4a, b and Supplementary Discussion). The modulus-switching mechanism of most traditional materials is based on the crystallization-melting transition or the glass transition. They tend to soften rather than stiffen upon heating, which was opposite to MALH, and the stimulus induced stiffening mechanism in fish skin or other stress reaction in biological system. Furthermore, most of soft robots were usually in soft state at room temperature and stiffen when needed. Therefore, MALH exhibited potential applications in soft robots. More importantly, different from the traditional thermal-responsive hydrogel (PNIPAAm) with a dramatic volume ratio change before and after phase separation (<0.1)[43], MALH can

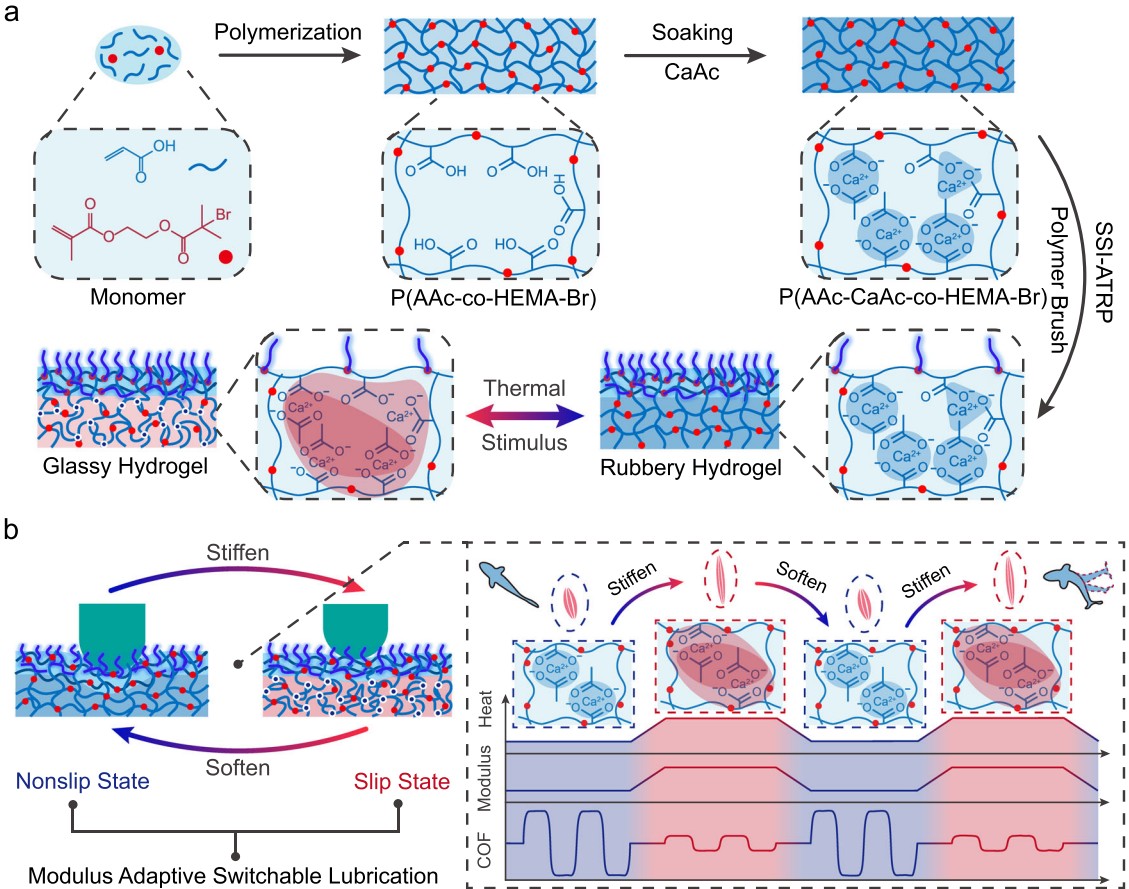

**Fig. 2 Preparation and modulus switch mechanism of MALH. a** Schematic illustration for the preparation of MALH and thermal-triggered modulus change mechanism. The dark blue filled area represented ionic interaction, the dark red and red filled area represented the cooperative effects of hydrophobic interaction and the enhanced ionic interaction. **b** The schematic diagram showing the fish-like adaptive switchable lubrication of MALH.

maintain the material geometry. The volume change ratio for all MALHs before and after phase separation were higher than 0.9 (Fig. 3e).

Furthermore, in order to measure the hardening temperature ($T_h$), the dynamic mechanical behavior of MALH was investigated during a temperature sweep (Fig. 3f). An abrupt increase in storage modulus ($G'$) and a peak in tanδ were observed at 46 °C in the rheogram, indicating a rubbery-to-glassy transition. Subsequently, a two-order-of-magnitude increase in $G'$ was observed when temperature increased continuously above $T_h$, indicating more rigid network formed gradually. This result was consistent with the increasing strength and modulus of MALH in strain-stress curves over 50 °C (Fig. 3b), which can be attributed to the hydrophobic interactions in $CH_3COO^-$ continuously enhancing the electrostatic interactions between $Ca^{2+}$ and $COO^-$ at high temperature[37,39,41]. It was also found that the content change of ATRP initiator had little effect on the $T_h$ (Fig. 3g). The differential scanning calorimetry (DSC) analysis was also used to investigate the phase-separation process (Supplementary Fig. 5). Finally, cyclic testing experiments (switch from 20 to 80 °C) indicated that the switchable mechanical properties of MALH based on phase separation were stable and reversible (Fig. 3h).

**Responsive lubrication of MALH.** The switchable characteristics of MALH between rigid and soft state enables the responsive lubrication property. Correspondingly, the thickness and microstructures of surface composite lubricating layer would also affect the interface friction performance[7,13,15]. Compared to P(AAc-

CaAc-co-HEMA-Br) hydrogel (Fig. 4a-i), the surface of MALH exhibited regular wrinkle patterns with clear gaps (Fig. 4a-ii, iii). The appearance of patterns can be attributed to the high grafting density of PSPMA. These wrinkles were highly hydrated after achieving swelling equilibrium in water (Fig. 4a-iv). The cross-section characterization showed that the lubricating layer was perfectly bonded onto the sub-surface of the phase-separation hydrogel (Fig. 4a-v). The lubricating layer thickness increased with extending the grafting time of PSPMA polyelectrolyte brush, while it leveled off after 30 min reaction (swelling: ~210 μm; deswelling: ~160 μm) (Fig. 4b). Detailed morphologies information for MALH samples with different grafting time were shown in Supplementary Figs. 6–8.

Subsequently, the responsive lubrication of MALH was investigated in water by employing a typical sphere-to-disk contact mode with reciprocating sliding test. As shown in Fig. 4c, the P(AAc-CaAc-co-HEMA-Br) hydrogel exhibited high COF (~0.5) at 20 °C (soft state) and low COF (~0.08) at 80 °C (rigid state). After surface grafting PSPMA brush as lubricating layer, the COFs of MALH for both states decreased significantly. At soft state (20 °C), the COFs decreased with extending the thickness of lubricating layer, mainly caused by the improvement of surface hydration capacity based on the water lubrication mechanism[10,44]. By contrast, the COFs of MALH increased slightly with extending the thickness of lubricating layer at rigid state (80 °C), resulting from the rising contact area and viscous force[7]. These results showed that the sharp modulus increasing of MALH can generate a dramatic decrease of one order of magnitude in COF. This phenomenon can be explained by the

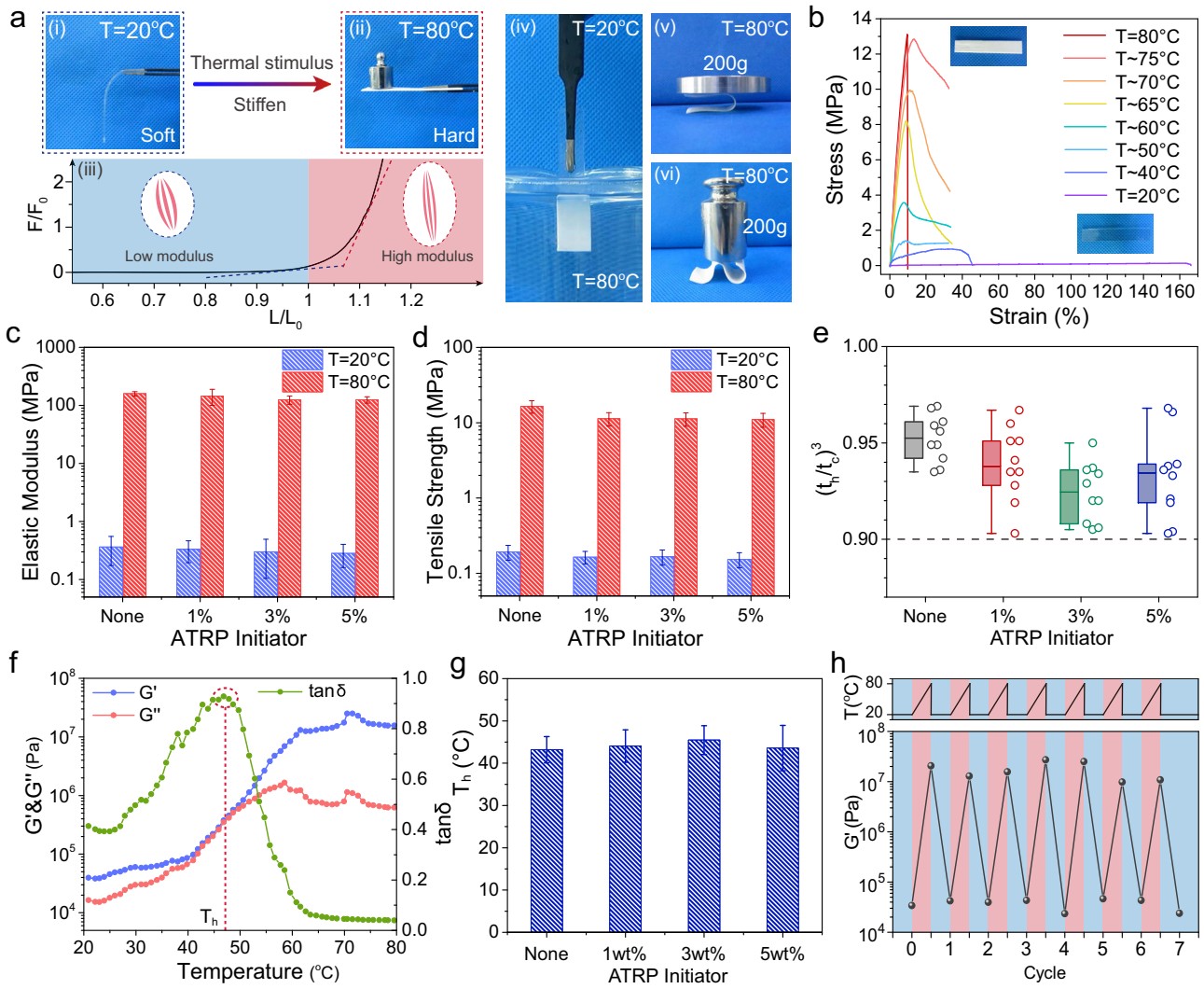

**Fig. 3 Switchable mechanical state of MALH in response to temperature. a** Demonstration for the phase-separation dominated mechanical characteristics of MALH at soft sate (20 °C) and rigid state (80 °C) (i, ii, iv), (iii) the schematic J-shaped curve of strain-stiffening mechanism of muscle ($L$ was the tensile length, $L_O$ was the rest length of muscle, $F$ was the tensile force, $F_O$ was the maximum isometric force)[35], (v, vi) demonstration for the load-bearing and shape-fixing capacity of MALH at rigid state. **b** Tensile test results of hydrogels with 3 wt% initiator from 20 to 80 °C. **c** The effect of the content of HEMA-Br (ATRP initiator) on elastic modulus and **d** tensile strength of MALH, the data were presented as mean ± SD and sample size $n = 6$. **e** Ratio of MALH volume change before and after phase separation, sample size $n = 10$ ($t_c$ and $t_h$ were the length before and after MALH phase separation, respectively; $t_h/t_c$ was the one-dimensional length change ratio, $(t_h/t_c)^3$ was the volume change ratio.). **f** Temperature-sweep rheological profile of MALH with 3 wt% ATRP initiator. **g** Phase-separation temperature ($T_h$) of MALH with different ATRP initiator content, the data were presented as mean ± SD and sample size $n = 8$. **h** Cyclic tests demonstrating reversible phase-separation behavior of MALH with 3 wt% ATRP initiator.

evolution of interface contact state. According to the two-term non-interacting model, friction force ($F$), was the sum of interfacial friction ($F_{int}$), and deformation component ($F_{def}$)[33,45]:

$$F = F_{int} + F_{def} \qquad (1)$$

Through a simple analysis of contact state in Hertz model (Fig. 4d), it was found MALH presented large contact radius (large contact area) and small contact stress in soft state (blue curve), while small contact radius (small contact area) and large contact stress in rigid state (red curve). As a result, the reduction of real contact area (~10.173 mm² to ~0.441 mm²) and the non-dissipation of contact stress (~0.087 MPa to ~2.983 MPa) collaboratively lead to a sharp decrease in $F_{int}$.

According to the result of nano-indentation test (Supplementary Fig. 9a, b), the modulus of lubricating layer in wet state was about 0.06 MPa, which was too weak to bear loads compared with the modulus of the P(AAc-CaAc-co-HEMA-Br) hydrogel.

Therefore, the load-bearing and deformation capacity of MALH was mainly dominated by the P(AAc-CaAc-co-HEMA-Br) substrate. The hardening of the substrate after phase separation would greatly reduce the deformation dissipation ($F_{def}$) during friction. The deformation of rigid MALH under loading was much smaller than that of soft MALH in contact simulation. Correspondingly, the results from macroscopic indentation tests (sphere indentor, same as friction test) also proved that rigid MALH was deformed less than that of soft state (Supplementary Fig. 10a). Both the experiment results and simple simulations demonstrated the effect of deformation dissipation on friction. Therefore, the modulus adaptive strategy can reduce interface friction force and then improve the local lubricating performance by controlling the interface contact state. Taking the lubrication performances of soft and rigid states into consideration, MALH-30 min sample (soft state: $\mu \sim 0.37$; rigid state: $\mu \sim 0.027$) was selected for further experiment.

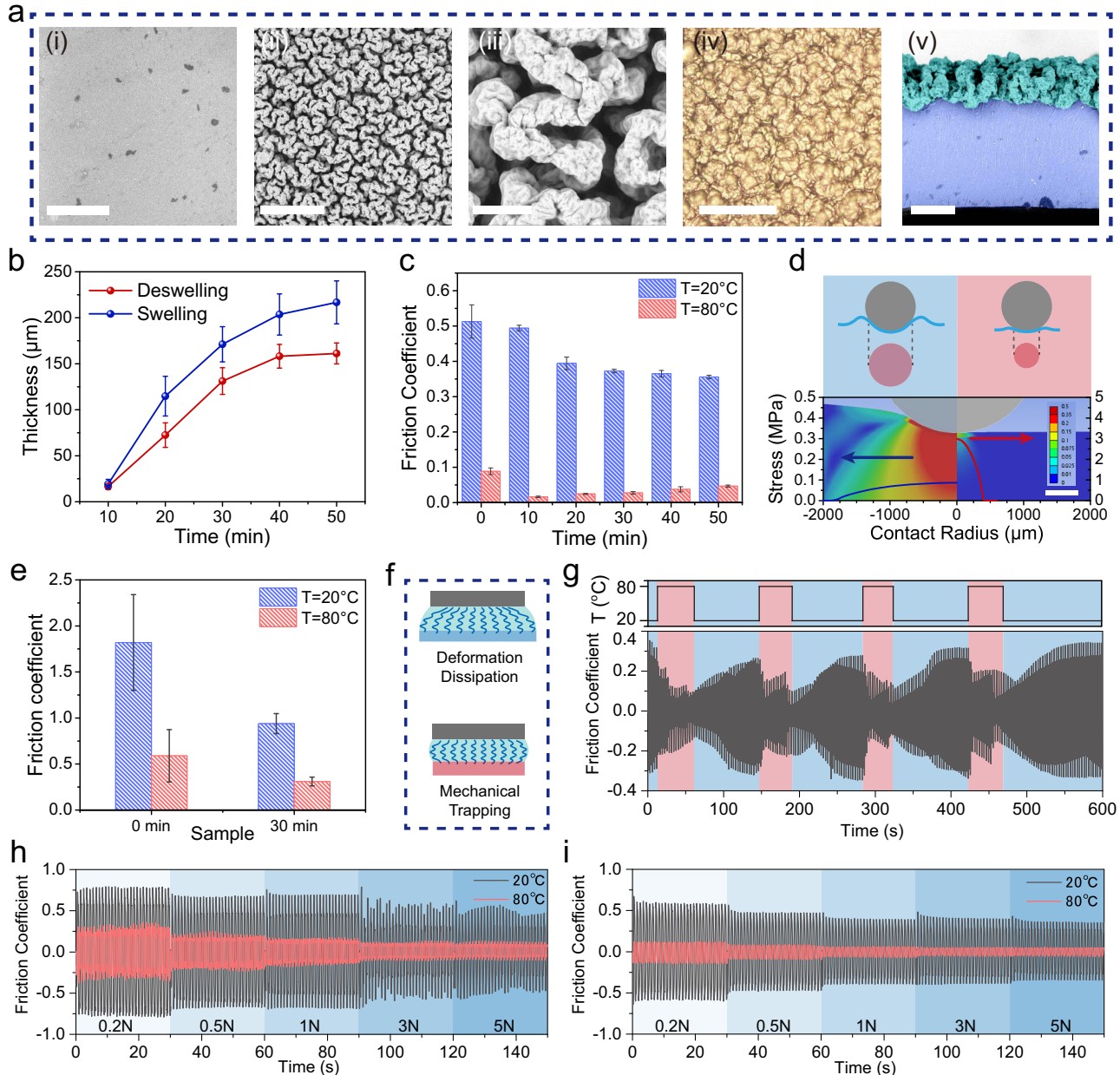

**Fig. 4 Responsive lubrication behavior of MALH. a** SEM images showing the dry surface morphology of (i) P(AAc-CaAc-co-HEMA-Br) hydrogel (scale bar: 300 μm), (ii)-(iii) MALH with different grafting time (scale bar: 300 and 50 μm), (iv) optical microscope photo showing the wet surface morphology of MALH (scale bar: 200 μm) and (v) SEM image showing the cross-section morphology of MALH (scale bar: 200 μm). **b** The relationship between lubricating layer thickness and grafting time of polymer brush of MALH in deswelling state and swelling state, the data were presented as mean ± SD and sample size $n = 5$. **c** Friction coefficient of MALH samples with different lubricating layer thickness at soft state (20 °C) and rigid state (80 °C), the data were presented as mean ± SD and sample size $n = 3$. **d** Schematic diagram showing the contact area of MALH against glass ball, and corresponding interface contact stress analysis result at soft state (blue curve) and rigid state (red curve) and the finite element analysis of deformation. Scale bar 1.25 mm. **e** Friction coefficient of P(AAc-CaAc-co-HEMA-Br) hydrogel and MALH (grafting time: 30 min) tested by a typical disk-to-disk contact mode, the data were presented as mean ± SD and sample size $n = 3$. **f** Mechanical trapping mechanism of MALH. **g** Evolution of COF by in-situ heating and cooling MALH (grafting time: 30 min). **h, i** COF signals for P(AAc-CaAc-co-HEMA-Br) hydrogel and MALH (grafting time: 30 min) at soft and rigid state, along with increasing the loads.

To confirm the contribution of the non-dissipative contact and mechanical trapping mechanism of polymer brush to lubrication, a disk-to-disk friction testing mode was used to eliminate the effect of the real contact area and minimize the elastic deformation. As shown in Fig. 4e, the COF of MALH can still decrease by ~3 times after phase separation. According to the contact mechanics theory[21], the instantaneous increase of substrate's modulus can induce in-situ mechanical trapping of

polymer brush (Fig. 4f), for which presents an important role in lubrication[8]. This phenomenon can be reflected by the increase of load and penetration depth in nano-indentation test (Supplementary Fig. 9c, d). After the phase separation, the measured surface modulus will be higher than before due to the substrate effect and the mechanical trapping effect. This trapping mechanism can be further confirmed by employing a disk-to-disk macroscopic indentation test. In this test, the interface

contact area was the same, and the deformation difference of substrates at rigid and soft states can be almost negligible (Supplementary Fig. 10b). The difference of the density and modulus of the polymer brush before and after mechanical trapping resulted in the different lubrication state. These obtained mechanisms (contact state evolution, non-deformation dissipation and mechanical trapping) also explained the switchable lubrication that occurs when fish struggling.

After friction testing, the surfaces of MALH samples with different lubricating layer thickness were observed by optical microscope (Supplementary Fig. 11). The P(AAc-CaAc-co-HEMA-Br) hydrogel showed apparent surface wear both at soft state and rigid state. By contrast, MALH exhibited weak or ignorable wear due to the presence of surface lubricating layer. Meanwhile, slight wear appeared for MALH sample with 10 min polymer brush grafting at soft state, resulting from its poor load-bearing capacity and relative thin thickness of lubricating layer. For MALH samples with 20 min to 50 min polymer brush grafting time, almost no surface wear were observed both at soft and rigid states. Although the weak load-bearing capacity in soft state and the high contact stress in rigid state can cause wear easily, the lubricating layer can effectively dissipate contact stress to prevent MALH from wearing[7].

By in-situ heating and cooling MALH to induce its modulus change, the interface COF can switch rapidly and reversibly (Fig. 4g). This result suggested that the current material system can mimic the switchable lubrication phenomenon that occurs during fish struggling. To verify the stability of MALH under a harsher mechanical loading environment, in-situ loading testing was processed. Meanwhile, the P(AAc-CaAc-co-HEMA-Br) hydrogel was used as a comparison. COFs signals of both blank hydrogel (Fig. 4h) and MALH (Fig. 4i) at soft state and rigid state exhibited friction-reduction phenomena with increasing the normal loads from 0.2 N to 5 N, implying a typically mechanical loading adaptive lubrication feature. Meanwhile, MALH presented more stable friction signals and lower COF at rigid state under wide range of loads, indicating the robustness of lubricating layer. This result was consistent with the fact that large grasping forces lead to being more difficulty to catch fish.

**Demonstration of MALH as intelligent bullets.** The unique lubrication-switching ability of MALH enabled its potential application as one kind of intelligent puncture device. As a demonstration, MALH was fabricated as a bullet with the assistance of mold (Fig. 5a). To demonstrate the functionality of MALH bullet more intuitively, a home-made pressure ejector was used as a pistol to shoot the MALH bullet (HB-L) with input energy of ~30 J, and a plasticine matrix was acted as base target (Fig. 5b). Meanwhile, the P(AAc-CaAc-co-HEMA-Br) hydrogel bullet (HB) was also used as a comparison. The bullets at soft state and rigid state would eventually stop upon encountering resistance from plasticine. After shooting, the corresponding piercing depths of bullets at soft state and rigid state were measured. Given the same input energy, the depth of the bullet was determined by friction force. Figure 5c showed the photo of the shooting experiment and the impacting surface of plasticine before and after shooting. The bullet hole left in the plasticine indicated that the bullet had been successfully shot inside (Supplementary Movie 2). It was found that both HB-L and blank HB presented a larger depth at rigid state (80 °C) than their soft state (20 °C) (Fig. 5d). HB-L exhibited larger depths than that of HB both at rigid and soft state (Fig. 5e). These results indicated that the synergistic contribution from mechanical adaptive modulus change and surface lubrication were of vital importance for switchable lubrication.

**Fabrication and demonstration of MASLD.** Although the conceptual display of the switchable lubrication behavior was successfully realized, it was still difficult to make MALH become an assembled device for more universal application. The existing hydration layer at surface of hydrogels can limit their adhesion ability and prevent them from attaching to other substrates[46,47]. In order to overcome this drawback, gradient cross-linked polydimethylsiloxane (PDMS) was covalently bonded onto one side of MALH to avoid the hydration layer and provide strong adhesion. As a result, one kind of modulus adaptive switchable lubrication device (MASLD) was prepared successfully. The cross-section morphology showed that MASLD was a four-layer device including lubricating layer (i), phase-separation hydrogel matrix (ii), highly cross-linked PDMS layer (iii), and lightly cross-linked PDMS layer (iv) (Fig. 6a).

The interface bonding strength between phase-separation hydrogel matrix and gradient cross-linked PDMS layer was investigated by employing a 90 ° peeling test. As shown in Fig. 6b, the hydrogel layer broke preferentially instead of interface failure, along with the considerable fracture strength of ~160 N/m, indicating that the real interface bonding strength was much larger than the hydrogel fracture energy[48]. The adhesion force of PDMS with different cross-linked ratios were investigated to optimize MASLD. The schematic diagram for normal adhesion testing process and typical adhesion forces-time curve of MASLD were presented in detail (Supplementary Fig. 12). The lower the cross-linked ratio was, the greater the adhesive force was, but the worse the mechanical strength was. So, highly cross-linked PDMS layer (20:1 curing ratio) was used as a bonding layer, while the lightly cross-linked PDMS layer (50:1) was used as adhesion layer. Based on the strong adhesion of gradient cross-linked PDMS layers[49], the MASLD could be used as a flexible patch to attach onto a series of substrates (Fig. 6c). In addition, the reversibility and water resistance ability of MASLD were investigated (Supplementary Fig. 13). No significant decrease of adhesion force was observed after encountering 20 times of repeated use or immersing into water for 7 days. As analyzed, the average adhesion force (~100 N/m) was much less than the interface bonding force (~160 N/m) (Fig. 5b), so it can be speculated that the interface failure of MASLD would not occur during repeated use process.

In order to demonstrate the modulus adaptive lubrication switching of MASLD, two pieces of MASLD patches were attached onto each side of the glass sheet. Subsequently, the MASLD modified glass sheet was captured by a mechanical arm (as a loading system, Fig. 6d) and pulled by a force sensor to obtain the instant interface friction force signal at 20 °C and 80 °C (Fig. 6e and Supplementary Movie 3). The applying constant loading force of per arm was calibrated and stabilized at about 400 mN in all cases (Supplementary Fig. 14). Instant force signals in pulling process indicated that friction force of blank glass sheet (~270 mN) was higher and more unstable than that of MASLD modified sample (~120 mN) at 20 °C. Furthermore, the modified sample exhibited lower friction force (~33 mN) after phase separation at 80 °C (Fig. 6f). Owing to the wide range of friction regulation (almost an order of magnitude), the MASLD can be applied to control the crawling of surface objects. As a demonstration, a MASLD patch was attached onto the surface of tilted glass plate with a slope of 45°, and then a tank was placed onto its surface (Fig. 6g). At low temperature (20 °C), due to the high interface friction force between the tank track and the MASLD patch, the tank can climb up the tilted plate. After heating up the patch to 80 °C, it was difficult for the mobile tank to climb up because of the modulus adaptive lubrication (Supplementary Movie 4).

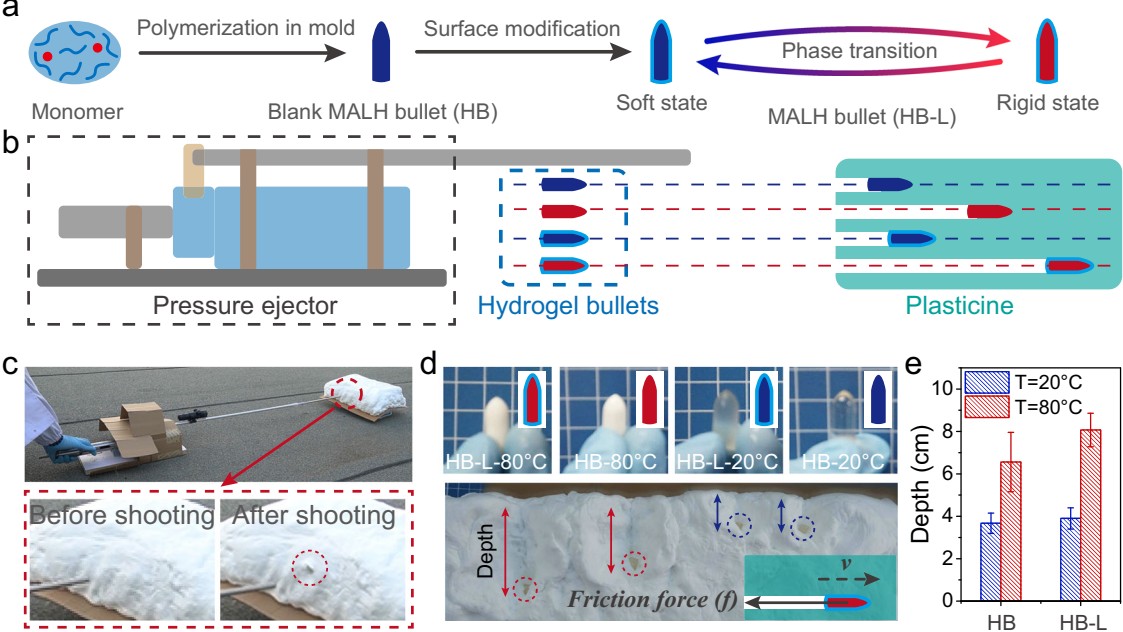

**Fig. 5 Demonstration of MALH as intelligent bullets. a** Schematic diagram showing the preparation process of MALH bullet. **b** Schematic diagram showing the shooting experiment at rigid (80 °C) and soft state (20 °C): a home-made a pressure ejector was used as a pistol to shoot the MALH bullet with input energy of ~30 J while the plasticine matrix was acted as base target. **c** The photo of the shooting ejector and the plasticine before and after shooting. **d** Optical photos showing the stagnant position of the HB-L and HB at rigid (80 °C) and soft state (20 °C). **e** The data of penetration depth. The data were presented as mean ± SD and sample size $n = 6$.

Furthermore, an artificial arm and a submarine were used to set up an underwater mechanical system to construct an in-situ capture device for demonstrating that MASLD can regulate friction force like a fish (Fig. 6h). The adhesive layer of MASLD patch can firmly attach onto a submarine with ~16 mN driving force, where the artificial arms were used as a constant loading capture system (~200 mN). As shown in Fig. 6i, the moving submarine could be easily captured by mechanical arms at 20 °C water bath because of high interface friction force. By contrast, the submarine instantly escaped at 80 °C, resulting from the low interface friction force of the MASLD after phase separation. The details of this demonstration experiment were provided in Supplementary Movie 5. At present, most of the bionic fish robots can realize the on-demand control of their soft and rigid moving states by manipulating the mobile joints, so as to change the flow field to achieve efficient drag-reduction[50]. Different from these bionic fish robots relying on change of mechanical movement states, MASLD can achieve drag-reduction through a collaborative strategy of coupling adaptive modulus with surface lubrication. Moreover, this strategy was based on mimicking the dynamic contact state between fish and objects. These demonstrations mimicked the escape situation and lubrication enhancement mechanism of fish, and indicated that the MASLD patch can be used as an intelligent switchable strategy for controlling the movement of mobile devices.

## Discussion

Inspired by the switchable lubrication feature of *longsnout catfish* from instant muscle-hardening mechanism, modulus adaptive lubricating hydrogel (MALH) was successfully developed. MALH could realize reversible mechanical state transition from a soft state (~0.3 MPa) at low temperature to a rigid state (~120 MPa) at high temperature based on the phase separation of hydrogel matrix. Such mechanical state transition of MALH endowed its switchable surface state from high friction state ($\mu$~0.37) to super-lubricating

state ($\mu$~0.027) in response to temperature. Such responsive lubrication feature was highly effective for wide range of normal loads, which can be explained by the synergy from contact state evolution, non-deformation dissipation and mechanical trapping mechanism. As an intuitive demonstration of its application potentials, MALH bullet was successfully prepared and exhibited excellent lubrication switch ability. Finally, a modulus adaptive switchable lubrication device (MASLD) was developed as a portable patch to apply in mobile devices for controlling their movement. This work provided an inspiration for designing intelligent lubrication system and soft robots.

## Methods

**Synthesis of the SSI-ATRP initiator**. The SSI-ATRP initiator was synthesized from a previously described procedure with slight modifications[7]. A mixture solution of HEMA (11.45 g, 0.088 mol) and triethylamine (4.05 g, 0.04 mol) in dichloromethane (45 g, 0.53 mol) in a flame dried round-bottom flask was cooled to 0 °C in an ice bath under nitrogen atmosphere for 20 min. Subsequently, 2-bromoisobutyryl bromide (18.393 g, 0.08 mol) in dichloromethane (23.25 g, 0.274 mol) was added dropwise to the mixture solution. The resultant solution was stirred at 0 °C for 4 h to complete the reaction. The white precipitate was filtered off and washed with dichloromethane twice. The liquid was collected and washed, respectively, with saturated $Na_2CO_3$ and saturated NaCl for several times until the solution pH was at about 7. Finally, the solvent was evaporated and the crude product was purified by column chromatography, resulting in a pale yellow liquid. The yield of this initiator is 60%. The chemical structure of the product was characterized by $^1$H-NMR spectrometer (Bruker, 400 MHz) and dimethyl sulfoxide (DMSO-d6) was used as solvent: 1.8-1.95 (m, 9H, $CH_3$), 4.35-4.42 (m, 4H, O-$CH_2$), 5.69 (s, 1H, C=$CH_2$), 6.05(s, 1H, C=$CH_2$).

**Preparation of composite hydrogel**. The composite PAAc hydrogel with SSI-ATRP initiator was synthesized via two steps. First, monomer of AAc (2.16 g, 30 mmol), initiator of α-Ketoglutaric acid (10 mg, 0.068 mmol), crosslinker of MBAA (23 mg, 0.15 mmol), HEMA-Br initiator with different mass ratios (0%, 1%, 3%, 5% mass ratios of AAc) were added in 10 mL pure water to obtain homogeneous solution. After being vigorously stirred for 10 min with gassing nitrogen, the resulting solution was subjected to UV irradiation for photo-initiators for 4 h to form a covalently cross-linked hydrogel. Then, the covalently cross-linked hydrogel was immersed in the CaAc solution (250 mM) for 2 weeks to form the phase-separation hydrogel.

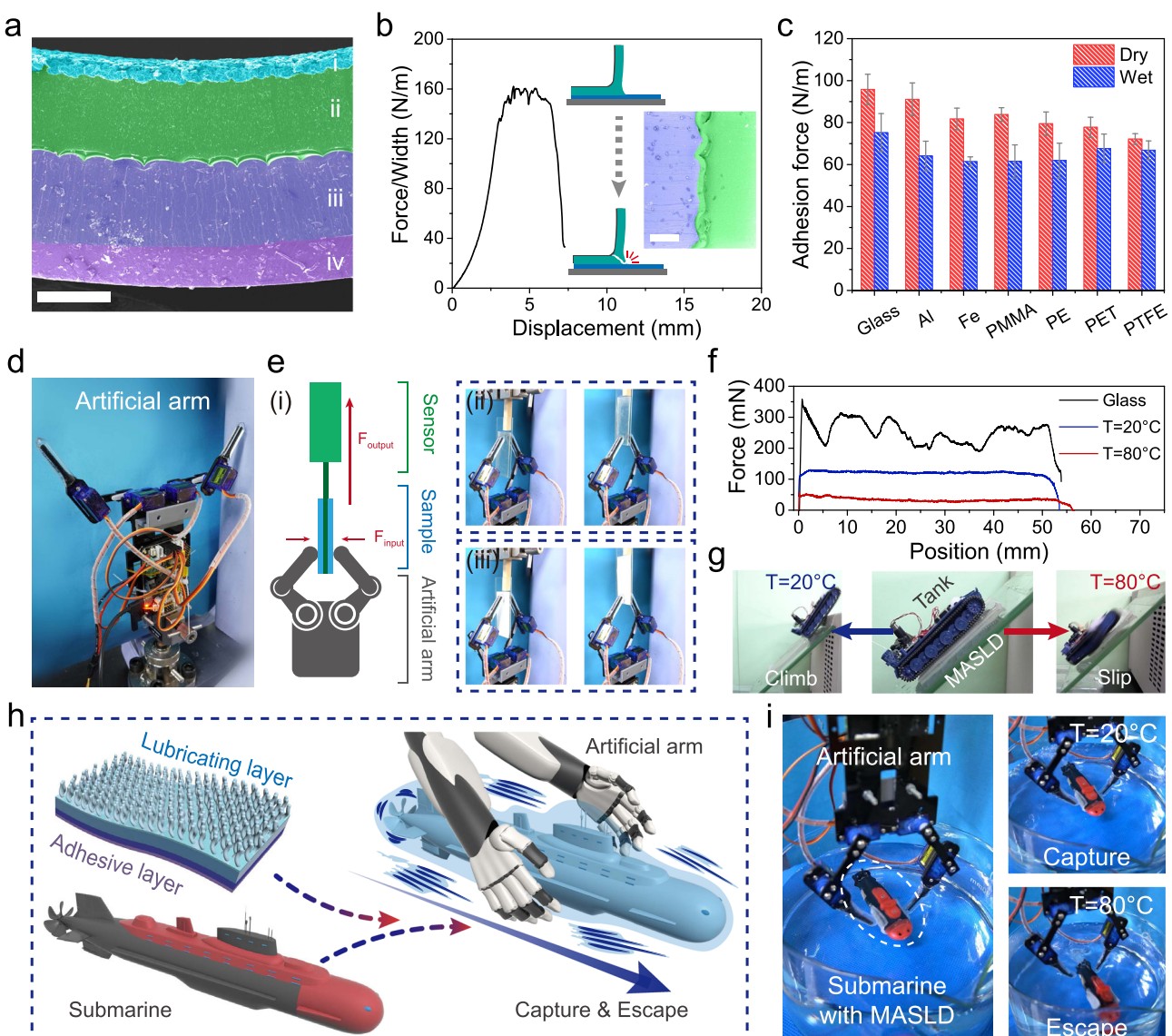

**Fig. 6 Adhesion, lubrication performance, and demonstration of MASLD. a** The cross-section SEM morphology of MASLD including lubricating layer (blue, i), phase-separation hydrogel matrix (green, ii), highly cross-linked PDMS layer (indigo blue, iii) and lightly cross-linked PDMS layer (purple, iv), scale bar: 500 μm. **b** The 90° peeling test for evaluating the interface bonding strength between the PDMS and hydrogel matrix, and the fracture diagram during test. The insert was enlarged SEM image of interface, scale bar: 300 μm. **c** Adhesion behavior of MASLD for different substrates, the data were presented as mean ± SD and sample size $n = 3$. **d** Artificial arms used in force testing and capture system. **e** Schematic diagram to demonstrate the testing process for responsive lubrication behavior of MASLD by applying artificially mechanical arms (e-i). Two pieces of MASLD patches were attached onto each side of the glass sheet, which was linked with a force sensor, and then interface friction force signals can be captured by pulling at 20 °C (e-ii) and 80 °C (e-iii). PDMS-decorated mechanical arms were used as loading systems. **f** Force signals in pulling process between the samples and arms. **g** Controllable movement of a mobile device on a MASLD-decorated tilted glass plate in response to temperature. **h, i** Schematic diagram and demonstration of the underwater mechanical system to in-situ capture device with MASLD, and the device escaped like fish.

**Preparation of gradient cross-linked PDMS**. Different ratios of the PDMS and curing agent (10:1, 20:1, 30:1,40:1, 50:1) were well mixed in a plastic cup and vacuumed in a vacuum pump for 20 min to remove air bubbles. Non-gradient cross-linked PDMS was cured at 80 °C for 4 h to make it cured. For gradient cross-linked PDMS (50-20:1), PDMS (20:1) was cured at 80 °C for 15 min to make it partially cured. Then, the lightly cross-linked PDMS precursor (50:1) was poured on top of the precured 20:1 PDMS, and the two components were subsequently cured at 80 °C for 4 h to make them completely joined together through an interfacial hydrosilylation reaction.

**Bonding composite hydrogels on gradient cross-linked PDMS**. The gradient cross-linked PDMS surfaces were thoroughly cleaned with deionized water and methanol, and completely dried with nitrogen gas. Thereafter, benzophenone solution (10 wt.% in ethanol) was applied onto the film to evenly cover the 20:1 surface for 5 min. The 50:1 surface was protected by 3 M tape. Then, the film

was washed with methanol and completely dried with nitrogen gas. To enhance wettability of the film, the film was treated by oxygen plasma for 15 s. The plasma treated side was then immersed into hydrogel monomer aqueous solution mentioned above. The total system was subjected to UV irradiation for photo-initiators for 4 h. After formation of hydrogel layer, the film was immersed in the CaAc solution (250 mM) for 2 weeks to form the phase transition hydrogel. As part of the sample preparation process, half of the film of some sample was masked with adhesive tape to prevent chemical bonding to the PDMS (20:1) substrate, which creates the arm for the 90° peeling test.

**Synthesis of PSPMA brush-grafted hydrogels**. Polymer brushes were grafted from initiator embedded composite hydrogel substrate by ATRP technique. The typical polymerization process was as follows. The monomer, SPMA (6 g, 24.4 mmol) was dissolved in the 8 ml water and 4 ml methanol at room temperature and degassed for 30 min with $N_2$. And then, the 2,2'-Bipyridine (80 mg,

0.512 mmol) and CuBr (35 mg, 0.244 mmol) were added in this solution. The mixture was further stirred and degassed with $N_2$ for another 20 min. The composite hydrogel sheet with 3 M tape protected on side and composite hydrogel with PDMS was put into the reaction solution under $N_2$ protection without being stirred for a certain period (10, 20, 30, 40, and 50 min). Finally, the samples were taken out and washed with deionized water to remove any unreacted monomers and catalysts.

**Mechanical properties measurement**. The mechanical properties were measured using an electrical universal material testing machine with a 500 N load cell (EZ-Test, SHIMADZU). The testing velocity was kept at 20 mm/min for tensile measurement and 50 mm/min for 90° peeling test. For tensile measurement, the sample was cut into sheets (size: $50 \times 5 \times 1$ mm). For 90° peeling test for the adhesion layer, the sample was cut into specific size ($100 \times 10 \times 2.5$ mm). For 90° peeling test for the interface bonding strength, the sample was cut into sheets (size: $70 \times 10 \times 3.5$ mm). A thin plastic film was placed behind the hydrogel to prevent it from deformation. Composite hydrogel was immersed in hot water ($T = 80\,°C$) for phase separation and waited for different times for the tensile test.

**Thermal shrinkage test**. The thermal response hydrogel was cut into long strips (size: $90 \times 4 \times 1$ mm, and then immersed in hot water ($T = 80\,°C$) for phase separation for performing this measurement. The sample length before phase separation was $t_c$, and the sample length after phase separation was $t_h$. The ratio $t_h/t_c$ was the one-dimensional length change ratio, and $(t_h/t_c)^3$ was the volume change ratio.

**Rheometry test**. The viscoelastic properties of the composite hydrogels with different initiator content were analyzed with rheometer (RS6000, HAAKE, Germany). A gel disc 35 mm in diameter was used for test. A small amount of 250 mM CaAc solution was added to the platform to prevent the sample from drying. Before starting the measurement, sample was pre-equilibrated at 20 °C for 30 min. Temperature sweep measurements of the storage ($G'$) and loss ($G''$) moduli were carried out from 20 to 80 °C at a heating rate of 3.0 °C/min, an angular velocity of 10 rad/s, 0.1% strain. For the rheological measurements, $T_h$ was defined as the temperature at which a peak was observed in a plot of tanδ vs. $T$, where tanδ was defined as $G''/G'$. To assess the reversibility of thermal hardening, the temperature was increased from 20 to 80 °C at a rate of 3 °C/min and lowered to 20 °C and held there for 5 h. This cycle was repeated seven times.

**DSC analysis**. DSC measurements were performed on a simultaneous thermal analyzer (NETZSCH STA 449F3). A small piece of gel (~2 mg) was encapsulated in a hermetic $Al_2O_3$ pan. The sample was pre-equilibrated at room temperature and then was heated at a rate of 10 °C/min, while the measurements were performed from 25 to 100 °C in air atmosphere. Phase-separation temperature, $T_h$, was defined as the maximum tangent of the incline due to the exothermic shift.

**Nano-indentation test**. Nano-indentation tests were performed on Bioindenter UNHT³ Bio (Anton Paar). The radius of the indenter (Ruby ball) was 0.5 mm. The samples were fixed on a flat glass slide by Loctite 496 glue in all tests. The thickness of lubricating layer was ~200 μm, normal load of 50 μN (displacement < 10 μm) was chosen to test the wet modulus of surface lubricating layer for eliminating the substrate effect. To detect the substrate effect and mechanical trapping effect, normal load of 500 μN (displacement > 10 μm) was chosen to test the samples.

**Macroscopic indentation test**. Macroscopic indentation tests were performed on adhesion testing machine (RTEC 2430, MFT-3000). The samples were fixed on a flat glass slide by Loctite 496 glue in all tests. For the sphere-to-disk test, a glass ball with a diameter of 6 mm was used as a contact pair, load =1 N. For the disk-to-disk test, one piece of glass (size: $10 \times 10 \times 1$ mm) was used as contact pair and the load was selected as 1 N. Limited by the instrument sensor, the total deformation/displacement measured includes the deformation of samples and strain sensor. The real deformation/displacement was the measured displacement minus the reference displacement. All the test conditions were the same as the friction test. Moreover, the sphere-to-disk result was used for a Hertz contact model analysis and finite element analysis.

**The morphology characterization**. The SEM (Phenom ProX G5) and optical microscope Olympus BX51 were employed to observe the surface morphology of samples. Moreover, for the polymer brush growth kinetics characterization, cross-section of the samples being immersed in water was imaged by optical microscope to observe the swelling thickness of polymer brushes, and cross-section of the samples dried was imaged by SEM to observe the deswelling thickness of polymer brushes.

**The friction characterization**. The friction test was performed on conventional sphere-on-disk tribometer by recording the friction coefficient under different conditions at 20 and 80 °C by using Tribometer (Anton Paar). For the sphere-to-disk test, a glass ball with a diameter of 6 mm was used as contact pair. In friction testing experiment for samples with different SSI-ATRP reaction time, the load was selected as 1 N. In in-situ friction coefficients testing experiment in response to

MALH heating and cooling, the load was also selected as 1 N to investigate the phase separation-induced switchable lubrication behavior. In order to perform loads-dependent switchable lubrication behavior of MALH, the loads were selected as 0.2, 0.5, 1, 3, and 5 N to evaluate its load-bearing capability, adaptability and stability. For the disk-to-disk test, one piece of glass (size:$10 \times 10 \times 1$ mm) was used as contact pair while the load was selected as 1 N. The MALH samples were fixed on a flat glass slide by Loctite 496 glue in all testing processes. Upon fixing the MALH samples as tested substrates, the overall surface of the samples must be maintained at a super-flat state, especially in the disk-to-disk contact test. The distance of one sliding cycle was 5 mm with a reciprocating mode and the friction coefficients were calculated by dividing the friction force by the applied normal load by the assistance of the software provided with the tribometer.

**Adhesion force measurement**. The adhesion forces were measured using adhesion testing machine (RTEC 2430, MFT-3000). In the wet experiments, the samples were submerged in water without vacuumed. In the underwater experiments, the samples were submerged entirely in water and vacuumed in a vacuum pump for at least 20 min to remove air bubbles at the interface. The sensor partially submerged in water was treated by plasma to avoid the capillary effect. Each test consisted of preloading, continuous loading, and pulling off. Measurement methods chose 40 s for preloading and 20 s for continuous load to achieve equilibrium. The pulling-off force was determined while withdrawing the glass sphere or flat sheet from the contact at a velocity of 0.1 mm/s.

**Shooting experiment**. The bullet was launched on a home-made platform powered by nitrogen. The launching pressure was 10 MPa. The bullet was placed in a seamless steel tube with a diameter of 8 mm and a thickness of 0.25 mm. The bullet accelerated at a distance of about 7 cm in tube. The launch exit was about 5 cm from the plasticine. The ambient temperature was 15 °C and humidity was 20%. After the bullet was shot into the plasticine, a knife was used to cut it open and measure the depth. Bullet diameter: 7.5 mm, length: 20 mm.

## Data availability
The data supporting the findings of this study are available within the paper, Supplementary Information, and Supplementary Movies. Source data are provided with this paper.

## Code availability
No custom code or mathematical algorithm is used in this manuscript.

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

## Acknowledgements

We are grateful for financial support from Key Research Program of the Chinese Academy of Sciences (XDPB24), National Natural Science Foundation of China (22032006 and 52075522), Outstanding Youth Fund of Gansu Province (21JR7RA095), and the Youth Innovation Promotion Association (2019411).

## Author contributions

Y.Z., S.M., and F.Z. conceived the idea. S.M. and Y.Z. designed the experimental protocol. Y.Z. performed and completed the entire experimental studies. W.Z., H.L., X.W., B.Y., M.C., and X.Z. provided technical suggestions. Y.Z., S.M., and F.Z. wrote the paper and all authors discussed the manuscript. S.M. and F.Z. supervised the entire research.

## Competing interests

The authors declare no competing interests.
