## [Peer Review File · Nature Communications]

Modulus adaptive lubricating prototype inspired by instant muscle hardening mechanism of catfish skinREVIEWER COMMENTS

Reviewer #1 (Remarks to the Author):

The authors describe a novel switchable lubrication system consisting of a hydrated polymer brush grafted on a hydrogel with tunable modulus. The coefficient friction can be tuned by one order of magnitude from 0.03 to 0.4 when the hydrogel is switched from its rigid to soft state (by temperature). The authors also demonstrate a way to apply the switchable lubricating hydrogel coatings on different substrates by using a PDMS backing.

The work is excellent and I recommend its publication in Nat. Comm. Novel concept. Well characterized system and very thorough work. I do have criticisms/comments that I hope the authors will address:

1) Language of the manuscript needs improvement. There are many unnecessary words, such as 'vividly', 'obviously'. Sentences should be made shorter and hence easier to read.

For example, take a look at the sentence

"As a proof of concept, our MALH can vividly mimick (sic) muscle evolution behavior of fish before and after contracting or tensing, so as to achieve the switchable lubricating phenomena"

This can be simplified to

"Our MALH can achieve switchable lubrication by mimicking the contraction and relaxation of fish muscles."

with no change in meaning. The word 'vividly' is ambiguous. Does it mean 'accurately', 'realistically'? One solution is to remove such adverbs (and other unnecessary adjectives).

There are many such examples in the text. Thankfully, the figures the authors created are excellent and clear. Hence, I am able to follow the paper easily. Improving the text will greatly increase the impact of the paper.

2) I do not like the bullet example given by the authors. It's not crucial and dilutes the content of the paper. It should be put in the Supporting information. Though, I accept this is a personal preference. I will leave the final decision to the authors and the editor.

3) 'th/tc' in Figure 3e not defined. Just change to volumetric change?

4) The authors used doubly charged Ca^{2+} ions in this paper. Will using different ions change the results significantly? I imagine using triply charged ions, e.g. Y^{3+} , will accentuate the change in modulus.

5) Fig. 4d is not clear. What does the black and white colouring means (in the plot)? And how did the authors calculate the stress? Did they use a Hertz contact model? Please elaborate.

6) Fig. 6a. I suggest indicating the different layers on the diagram with words. Not just in the caption. I (and a few other readers) don't know the difference between indigo blue and purple. Indicating the layers directly on the figure will solve this problem.

D. Daniel,

IMRE, A*STAR

Reviewer #2 (Remarks to the Author):

This manuscript describes a new class of modulus adaptive lubricating hydrogel (MALH) that is inspired by instant muscle hardening mechanism of catfish skin. The authors made an elegant design of MALH hydrogel consisting of top mucus-like hydrophilic lubricating layer and muscle-like bottom hydrogel that can be hardened from a soft state (with high friction state, $\mu \sim 0.37$) at low temperature to a rigid state (super-lubricating state, $\mu \sim 0.027$) at high temperature. The work is initiated from an important problem and designed with their novelty solution. In addition, the manuscript is attractive for its high quality experimental design and data presentation. The modulus adaptive lubricating hydrogel demonstrated in this manuscript should have great contribution to many fields such as motion devices.

Based on the evaluation of the novelty and quality of this manuscript, I strongly recommend acceptance of this manuscript for publication in Nature Communications after the authors address some minor points.

1. Although this is an excellent piece of work in terms of novelty, it is unnecessary to use words such as “novel” and “new” in the manuscript, such as line 18, 80, 276, 305, and 384.
2. The size of samples used for mechanical properties measurement is missing and should be defined.
3. For all bar graph, the n number should be defined in the relevant legend.
4. A reference related to the synthesis of the SSI-ATRP should be cited if available.
5. Related to point question 4, the description of the NMR result needs to be added at the end of the synthesis paragraph (line 42 in the Supplementary information). In addition, the frequency of the ¹H NMR characterization is also missing.
6. I would suggest adding characterization about the chemical component of P(AAc-co-HEMA-Br) hydrogel before and after brush-graft, such as ATR-FTIR or XPS.
7. Some codes have been repeatedly defined. For example, the author does not need to repeat the full name of the MALH in line 102, 113, and 514.
8. How was face-to-face test done in the Fig. 4e?
9. The experimental detail of DSC test is missing and should be added.
10. Please corrected the format mistake, including but not limited to: a comma symbol after Th should be removed in line 175; A space is required between the number and the unit, such as in line 46, 47, 57, 58, 59, 62...in the Supplementary information; Space before ref. should be removed in line 279, 289 in the Supplementary information. Font (°C) should be corrected in line 167 in the Supplementary information.

Reviewer #3 (Remarks to the Author):

The manuscript entitled "Modulus adaptive lubricating prototype inspired by instant muscle hardening mechanism of catfish skin" proposed a strategy for constructing mechanically adaptive layered hydrogels lubrication material by regulating substrate rigidity, as well as the contact state of interface. This concept well mimics the change of interface contact state caused by the skin system modulus switch in the process of fish movement, and the layered hydrogels lubrication material consists of top fish mucus-like hydrophilic lubricating layer and muscle-like bottom hydrogel that can be hardened via thermal-triggered phase separation. Impressively, it can realize as high as 14 times magnitude control of interface friction force in response to modulus change from soft gel state (0.3 MPa) to stiff plastic state (110 MPa) by thermal stimulus. Finally, as a proof of concept, an intelligent hydrogels bullets and a

swimming device integrated with this intelligent hydrogels material are designed, implying extraordinary application potential in military equipment and smart devices.

The authors reported a novel concept for realizing adaptively large-scale interface friction control, which is highly different from traditional surface hydration or viscoelastic state regulation strategies (passive control mean) (Science, 2018, 360(6396): 1434-1438, Matter, 2021, 4(2): 675-687). It is an important breakthrough in the field of surface/interface science and materials. The research was well designed. The description of experiments and discussion are all clear and sufficient. Correspondingly, the manuscript is well crafted and scholarly presented. It is of great interest for researchers working in the field of soft robots, interface intelligent devices, responsive polymers materials. I think this work can be published on top journal as Nature Communications. However, some minor issues need to be solved and some parts of the text should be further improved before publication.

- 1) The mechanism of this modulus adaptive lubrication is anchored in the contact area and mechanical deformation. The related experiment data in Figure S7a is not well explained or discussed in detail. The author should describe it in more detail in the main text.
- 2) In this paper, a face-to-face friction test was applied to MALH to ensure the interface contact area being the same and the deformation being negligible. The author thinks that the difference of COF mainly comes from the mechanical trapping mechanism of polymer brush. However, the related experiment data in Figure S7b to prove the interface contact state was not discussed in detail.
- 3) The authors compared the shrinkage rates of MALH and NIPAAM before and after phase separation. The author thinks that MALH system can well maintain the material geometry because of the slight volume shrinkage. However, the dramatic shrinkage of the volume will lead to an increase in the density of the polymer brushes on the surface, thus improving the lubrication performance. The author should give some explanation.
- 4) Necessarily, although it is difficult, the authors should provide the modulus data of the surface lubricating layer by micromechanics test.
- 5) This work compared MALH with other reported modulus-switching systems in supplementary section. The author should give more prominence to the characteristics of thermal stiffening.
- 6) MALH requires external thermal stimulus to achieve regulation. Especially, the authors used hot water as a stimulus. Can other stimuli or remote control be used to achieve regulation? Maybe this can broaden the range of application.
- 7) The comparison between MALSD and Ref.50 (Science Robotics, 2021, 6: eabe4088.) should be described in more detail. The fish-like robot in ref.50 improved the swimming efficiency by using a stiffness-tunable tail to control the flow field. In this work, the author needs to highlight the characteristics of the dynamic contact state evolution of the MALSD lead by the switchable modulus.

Response to reviewer #1:

General comment: *The authors describes a novel switchable lubrication system consisting of a hydrated polymer brush grafted on a hydrogel with tunable modulus. The coefficient friction can be tuned by one order of magnitude from 0.03 to 0.4 when the hydrogel is switched from its rigid to soft state (by temperature). The authors also demonstrate a way to apply the switchable lubricating hydrogel coatings on different substrates by using a PDMS backing. The work is excellent and I recommend its publication in Nat. Comm. Novel concept. Well characterized system and very thorough work.*

Response: Thank you for your insightful and constructive comments. To address yours and other reviewers' comments and concerns in full, we have added or revised figures, introduced clarifications, expanded discussions and improved the language in the revisions. In the following paragraphs, we address each comment point by point. Sentences newly added to or revised in the manuscript and the supplementary information are marked as in red.

Comment 1: *Language of the manuscript needs improvement. There are many unnecessary words, such as 'vividly', 'obviously'. Sentences should be made shorter and hence easier to read. For example, take a look at the sentence, "As a proof of concept, our MALH can vividly mimic (sic) muscle evolution behavior of fish before and after contracting or tensing, so as to achieve the switchable lubricating phenomena". This can be simplified to "Our MALH can achieve switchable lubrication by mimicking the contraction and relaxation of fish muscles." with no change in meaning. The word 'vividly' is ambiguous. Does it mean 'accurately', 'realistically'? One solution is to remove such adverbs (and other unnecessary adjectives). There are many such examples in the text. Thankfully, the figures the authors created are excellent and clear. Hence, I am able to follow the paper easily. Improving the text will greatly increase the impact of the paper.*

Response 1: We highly appreciate the reviewer's suggestion about the language improvement for our manuscript. We have deleted some ambiguous and unnecessary adverbs/adjectives, simplified some sentences to make our manuscript easier to read. These changes are marked in red in our revised manuscript.

Comment 2: *I do not like the bullet example given by the authors. It's not crucial and dilutes the content of the paper. It should be put in the Supporting information. Though, I accept this is a personal preference. I will leave the final decision to the authors and the editor.*

Response 2: Thank you for your comment regarding the bullet and shooting experiment in our manuscript. Maybe you thought the experiment is too “garish”. We want to demonstrate our lubrication strategy intuitively. During this experiment, the different penetration depths related to the friction force change is more direct than friction coefficient data and curves. So we decided to keep in our manuscript.

Comment 3: *' t_h/t_c ' in Figure 3e not defined. Just change to volumetric change?*

Response 3: Thank you for pointing out the issue in **Figure 3e**. We have defined the t_c , t_h and t_c/t_h in **Figure 3e** in the revised manuscript. Also, we have added detailed descriptions for this experiment in revised supplementary information.

In the revised manuscript:

Figure 3e. Ratio of the MALH volume change before and after phase separation. (t_c and t_h are the length before and after MALH phase separation, respectively; t_h/t_c is the one-dimensional length change ratio, $(t_h/t_c)^3$ is the volume change ratio.)

In revised supplementary information:

Thermal shrinkage test: The thermal response hydrogel was cut into long strips with size of 90 mm*4 mm*1 mm, and then immersed into hot water ($T=80\text{ }^\circ\text{C}$) to induce phase transition for performing this measurement. The sample length before phase separation is t_c , and the sample length after phase separation is t_h . The ratio t_h/t_c is the one-dimensional length change ratio, and $(t_h/t_c)^3$ is the volume change ratio.

Comment 4: *The authors used doubly charged Ca^{2+} ions in this paper. Will using different ions change the results significantly? I imagine using triply charged ions, e.g. Y^{3+} , will accentuate the change in modulus.*

Response 4: Thank you for raising this question. It is true that the different ions affect the property of this material. We actually conducted experiments to investigate the effect of different ions (M^+ , M^{2+} , M^{3+}) on modulus change. The results are shown in **Figure R1**.

- For monovalent metal ions, e.g. Na^+ , no obvious modulus change was found for P(AAc-NaAc-co-HEMA-Br) hydrogel with rising temperature from $20\text{ }^\circ\text{C}$ to $80\text{ }^\circ\text{C}$. And this hydrogel tended to swell.
- For trivalent metal ions, e.g. Al^{3+} or Fe^{3+} , P(AAc-AlAc-co-HEMA-Br) hydrogel and P(AAc-FeAc-co-HEMA-Br) hydrogel showed improved mechanical strength in comparison with other hydrogels at $20\text{ }^\circ\text{C}$, but there was still no obvious modulus change when rising temperature from $20\text{ }^\circ\text{C}$ to $80\text{ }^\circ\text{C}$.
- For divalent metal ions, e.g. Ca^{2+} or Zn^{2+} , we found phase separation phenomenon

and modulus change for P(AAc-CaAc-co-HEMA-Br) hydrogel and P(AAc-ZnAc-co-HEMA-Br) hydrogel.

The difference caused by different metal ions is attributed to their coordination ability. Fe^{3+} and Al^{3+} ions can directly crosslink the PAAc network due to its strong coordination ability with the carboxyl group, which resists the acetate residues penetrating into the hydrogel network. Na^+ ions cannot interact with carboxyl groups and acetate residues at the same time, and therefore has no modulus change effect.

Ca^{2+} and Zn^{2+} ions have moderate coordination ability and can interact with carboxyl groups and acetate residues at the same time, so they have modulus change effect. However, considering that high concentration and content of Zn^{2+} may be toxic, we used Ca^{2+} ions to engineer phase separation hydrogels in this work. Certainly, this does not affect the application potential of P(AAc-ZnAc-co-HEMA-Br) hydrogel in other fields. In fact, we are working on related research.

Figure R1: Modulus change of P(AAc-Mⁿ⁺Ac-co-HEMA-Br) hydrogel at different temperature (20°C and 80°C) and corresponding photos of samples.

Comment 5: Fig. 4d is not clear. What does the black and white colouring means (in the plot)? And how did the authors calculate the stress? Did they use a Hertz contact model? Please elaborate.

Response 5: Thank you for pointing out the issue in Fig. 4d. To make this figure clear, we change the black and white color into red and blue color. And we provided a new deformation result of MALH to replace the former result. Herein, we calculated the stress by using Hertz contact model. In the previous calculation, we neglected the influence of the surface lubricating layer on mechanics because this layer was thin and too soft to bear load, and the mechanical parameters of the surface lubricating layer was lacking. Therefore, our previous calculation results are higher than the actual contact stress. However, in response to the **reviewer #3's Comment 4**, we have tested the mechanical parameters of the surface lubricating layer in wet state by NHT³ Nanoindenter (Probe: Ruby ball; diameter: 1 mm). So, the lubricating layer has been taken into account in Hertz contact model. According to the force and displacement data in nano indentation test and the macroscopic contact indentation test, the surface lubrication layer is approximated as a gradient material, and a new contact stress and deformation are calculated (**Figure R2**). The new contact area for soft state and rigid state was $\sim 10.173 \text{ mm}^2$ and $\sim 0.441 \text{ mm}^2$, respectively. And the contact stress for soft state and rigid state was $\sim 0.087 \text{ MPa}$ and $\sim 2.983 \text{ MPa}$, respectively. Compared with the previous results without considering the soft lubrication layer (contact area for soft state and rigid state was $\sim 9.93 \text{ mm}^2$ and $\sim 0.194 \text{ mm}^2$, contact stress for soft state and rigid state was $\sim 0.15 \text{ MPa}$ and $\sim 7.73 \text{ MPa}$), the presence of soft surface lubricating layer would increase the interface contact area and reduce the contact stress. The calculation details are supplemented and provided in the supplementary information.

Figure R2 [Figure 4d]. (d) Schematic diagram showing the contact area of MALH against glass ball, and corresponding interface contact stress analysis result at soft state (blue curve) and rigid state (red curve) and the finite element analysis of deformation. Scale bar 1.25 mm.

Comment 6: *Fig. 6a. I suggest indicating the different layers on the diagram with words. Not just in the caption. I (and a few other readers) don't know the difference between indigo blue and purple. Indicating the layers directly on the figure will solve this problem.*

Response 6: Thanks for pointing this out. We have modified the diagram as required. We have indicated the different layers on the diagram with words: i, ii, iii, iv (**Figure R3**).

Figure R3 [Figure 6a]. The cross-section SEM morphology of MASLD including composite lubricating layer (blue, i), phase separation hydrogel matrix (green, ii), highly crosslinked PDMS layer (indigo blue, iii) and lightly crosslinked PDMS layer (purple,iv), scale bar: 500 μm .

III: Response to reviewer #2

General comment: *This manuscript describes a new class of modulus adaptive lubricating hydrogel (MALH) that is inspired by instant muscle hardening mechanism of catfish skin. The authors made an elegant design of MALH hydrogel consisting of top mucus-like hydrophilic lubricating layer and muscle-like bottom hydrogel that can be hardened from a soft state (with high friction state, $\mu \sim 0.37$) at low temperature to a rigid state (super-lubricating state, $\mu \sim 0.027$) at high temperature. The work is initiated from an important problem and designed with their novelty solution. In addition, the manuscript is attractive for its high quality experimental design and data presentation. The modulus adaptive lubricating hydrogel demonstrated in this manuscript should have great contribution to many fields such as motion devices. Based on the evaluation of the novelty and quality of this manuscript, I strongly recommend acceptance of this manuscript for publication in Nature Communications after the authors address some minor points.*

Response: Many thanks for your positive evaluation for novelty and results of our work.

Comment 1: *Although this is an excellent piece of work in terms of novelty, it is unnecessary to use words such as “novel” and “new” in the manuscript, such as line 18, 80, 276, 305, and 384.*

Response 1: We appreciate the reviewer’s suggestion. As a response, the words “new” and “novel” in the revised manuscript have been deleted.

Comment 2: *The size of samples used for mechanical properties measurement is missing and should be defined.*

Response 2: Thanks for your question. As a response, we have defined the size of samples used for mechanical properties test in the revised supplementary information.

Mechanical properties measurement: The mechanical properties were measured by using an electrical universal material testing machine with a 500 N load cell (EZ-Test, SHIMADZU). The testing velocity was kept at 20 mm/min for tensile measurement and 50 mm/min for 90° peeling test. For tensile measurement, the sample was cut into size of 50 mm*5 mm*1 mm. For 90° peeling test for the adhesion layer, the sample was cut into size of 100 mm*10 mm*2.5 mm. For 90° peeling test for the interface bonding strength, the sample was cut into size of 70 mm*10 mm*3.5 mm.

Comment 3: *For all bar graph, the n number should be defined in the relevant legend.*

Response 3: Thanks for your suggestion. As a response, now all the n number has been defined in the relevant legend of the bar graph.

Comment 4: *A reference related to the synthesis of the SSI-ATRP should be cited if available.*

Response 4: The reference related to the synthesis of the SSI-ATRP has been cited, the reference 1 in the supplementary information, and the reference 7 in the manuscript.

Comment 5: *Related to point question 4, the description of the NMR result needs to be added at the end of the synthesis paragraph (line 42 in the Supplementary information). In addition, the frequency of the ¹H NMR characterization is also missing.*

Response 5: Thank you for pointing out the issues in Supplementary information. As a response, the description of the NMR result and the frequency of the ¹H NMR characterization have been added at the end of the synthesis paragraph.

Synthesis of the SSI-ATRP initiator: ...The chemical structure of the product was characterized by ¹H NMR (400 MHz) analysis using Bruker 400 MHz NMR spectrometer and used dimethyl sulfoxide (DMSO-d₆) as solvent: 1.8-1.95 (m, 9H, CH₃), 4.35-4.42 (m, 4H, O-CH₂), 5.69 (s, 1H, C=CH₂), 6.05(s, 1H, C=CH₂).

Comment 6: *I would suggest adding characterization about the chemical component of P(AAc-co-HEMA-Br) hydrogel before and after brush-graft, such as ATR-FTIR or XPS.*

Response 6: Thanks for your valuable suggestion. We have added the ATR-FTIR characterizations about the chemical components of PAAc hydrogel, P(AAc-co-HEMA-Br) hydrogel, P(AAc-CaAc-co-HEMA-Br) hydrogel and the hydrogel grafting polymer brush, P(AAc-CaAc-co-HEMA-Br-PSPMA) (MALH) (**Figure R4a**). Adding HEMA-Br monomer to PAAc network made the peak at ~1710 cm⁻¹ broader than PAAc network. By soaking P(AAc-co-HEMA-Br) into CaAc solution, the CaAc binding peaks appeared in the gel network, and the P(AAc-CaAc-co-HEMA-Br) hydrogel network were stable after 2 weeks. After grafting PSPMA polymer brush on P(AAc-CaAc-co-HEMA-Br) hydrogel, the sulfo-group peak appeared at ~1050 cm⁻¹.

In addition, we also have added the characterizations about the surface chemical components of P(AAc-CaAc-co-HEMA-Br) hydrogel and P(AAc-CaAc-co-HEMA-Br-PSPMA) hydrogel (MALH) before and after phase separation (**Figure R4b**). It was found that the water and hydroxyl binding peak at ~3300 cm⁻¹ in P(AAc-CaAc-co-HEMA-Br) hydrogel decreased after phase separation. This is because the hydrophobic residues (acetate) can dehydrate at high temperature to form hydrophobic regions. For

P(AAc-CaAc-co-HEMA-Br-PSPMA) hydrogel (MALH), the water and hydroxyl binding peak of surface lubricating layer wasn't decreased. This means the surface lubricating layer of MALH still has good hydration ability and lubrication performance after phase separation. This reflects the advantage of multi-layer materials.

Figure R4 [Supplementary Figure 2]. (a) ATR-FTIR spectrum of CaAc, PAAc hydrogel, P(AAc-co-HEMA-Br) hydrogel, P(AAc-CaAc-co-HEMA-Br) hydrogel immersed in CaAc solution with different time, P(AAc-CaAc-co-HEMA-Br-PSPMA) hydrogel (MALH). (b) ATR-FTIR spectrum of P(AAc-CaAc-co-HEMA-Br) hydrogel and P(AAc-CaAc-co-HEMA-Br-PSPMA) hydrogel (MALH) before and after phase separation.

Comment 7: *Some codes have been repeatedly defined. For example, the author does not need to repeat the full name of the MALH in line 102, 113, and 514.*

Response 7: Thanks to reviewer for pointing out this issue. As a response, we have deleted and corrected these words defined repeatedly in this manuscript. We retained the full name of MALH when it first appeared in the abstract, introduction and discussion.

Comment 8: *How was face-to-face test done in the Fig. 4e?*

Response 8: Many thanks to your question. As a response, we have described more clearly both the testing conditions of sphere-to-face (disk) contact and the face-to-face (disk) contact in the revised supplementary information.

The Friction Characterization: ...For the sphere-to-disk test, a glass ball with a diameter of 6 mm was used as contact pair. In friction testing experiment for samples with different SSI-ATRP reaction time, the load was selected as 1 N. In *in-situ* friction coefficients testing experiment in response to MALH heating and cooling, the load was also selected as 1 N to investigate the phase separation induced switchable lubrication behavior. In order to perform loads-dependent switchable lubrication behavior of

MALH, the loads were selected as 0.2 N, 0.5 N, 1 N, 3 N and 5 N to evaluate its load-bearing capability, adaptability and stability. For the face-to-face test, one piece of glass (size: 10 mm*10 mm*1 mm) was used as contact pair while the load was selected as 1 N. The MALH samples were fixed on a flat glass slide by Loctite 496 glue in all testing processes. Upon fixing the MALH samples as a tested substrate, the overall surface of the samples must be maintained at a super-flat state, especially in the disk-to-disk contact test...

Comment 9: *The experimental detail of DSC test is missing and should be added.*

Response 9: Thank you for this careful review. As a response, we have added the experimental detail of DSC test in the supplementary information.

DSC analysis: DSC measurements were performed on a simultaneous thermal analyzer (NETZSCH STA 449F3). A small piece of gel (~ 2 mg) was encapsulated in a hermetic Al₂O₃ pan. The sample was pre-equilibrated at RT and then was heated at a rate of 10 °C/min, while the measurements were performed from 25 °C to 100 °C in air atmosphere. Phase separation temperature, T_h, was defined as the maximum tangent of the incline due to the exothermic shift.

Comment 10: *Please corrected the format mistake, including but not limited to: a comma symbol after Th should be removed in line 175; A space is required between the number and the unit, such as in line 46, 47, 57, 58, 59, 62...in the Supplementary information; Space before ref. should be removed in line 279, 289 in the Supplementary information. Font (°C) should be corrected in line 167 in the Supplementary information.*

Response 10: Thank you for pointing out these format mistakes in the manuscript. As a response, we have corrected them in the revised manuscript and supplementary information.

IV: Reviewer #3

General comment: *The manuscript entitled "Modulus adaptive lubricating prototype inspired by instant muscle hardening mechanism of catfish skin" proposed a strategy for constructing mechanically adaptive layered hydrogels lubrication material by regulating substrate rigidity, as well as the contact state of interface. This concept well mimics the change of interface contact state caused by the skin system modulus switch in the process of fish movement, and the layered hydrogels lubrication material consists of top fish mucus-like hydrophilic lubricating layer and muscle-like bottom hydrogel that can be hardened via thermal-triggered phase separation. Impressively, it can realize as high as 14 times magnitude control of interface friction force in response to modulus change from soft gel state (0.3 MPa) to stiff plastic state (110 MPa) by thermal stimulus. Finally, as a proof of concept, an intelligent hydrogels bullets and a swimming device integrated with this intelligent hydrogels material are designed, implying extraordinary application potential in military equipment and smart devices. The authors reported a novel concept for realizing adaptively large-scale interface friction control, which is highly different from traditional surface hydration or viscoelastic state regulation strategies (passive control mean) (Science, 2018, 360(6396): 1434-1438, Matter, 2021, 4(2): 675-687). It is an important breakthrough in the field of surface/interface science and materials. The research was well designed. The description of experiments and discussion are all clear and sufficient. Correspondingly, the manuscript is well crafted and scholarly presented. It is of great interest for researchers working in the field of soft robots, interface intelligent devices, responsive polymers materials. I think this work can be published on top journal as Nature Communications.*

Response: We extremely appreciate the reviewer's positive evaluation for concept novelty of our work, and recommend its publication in Nature Communications.

Comment 1: *The mechanism of this modulus adaptive lubrication is anchored in the contact area and mechanical deformation. The related experiment data in Figure S7a is not well explained or discussed in detail. The author should describe it in more detail in the main text.*

Response 1: We appreciate the reviewer's valuable suggestion. As a response, we have discussed the test data more detail in revised manuscript and supplementary information. Limited by the instrument sensor, the total deformation/displacement measured includes the deformation of MALH and strain sensor. So, we added the reference curve in Figure S7a and S7b (**Figure R5**). The real deformation/displacement is the measured displacement minus the reference displacement. The test conditions are

the same as the friction test. And this result was used for fitting in the Hertz contact model for finite element analysis.

Figure R5 [Supplementary Figure 10]. Indentation test of blank MALH without polymer brush and MALH at soft state (S) and rigid state (R) by using sphere indenter (a) and flat plane indenter (b). The reference curve was the sensor displacement under testing load. The reference displacement of sphere indentation and flat plane indentation was $488.8 \pm 5 \mu\text{m}$ and $439.8 \pm 13 \mu\text{m}$, respectively. Displacement of Blank-S in sphere indentation was $\sim 901 \mu\text{m}$, and that of MALH-S was $\sim 1017 \mu\text{m}$. Displacement of Blank-R in sphere indentation was $\sim 23 \mu\text{m}$, and that of MALH-R was $\sim 55 \mu\text{m}$. These data were fitted by Hertz model.

Comment 2: *In this paper, a face-to-face friction test was applied to MALH to ensure the interface contact area being the same and the deformation being negligible. The author thinks that the difference of COF mainly comes from the mechanical trapping mechanism of polymer brush. However, the related experiment data in Figure S7b to prove the interface contact state was not discussed in detail.*

Response 2: We appreciate the reviewer's suggestion about the discussion improvement of interface contact state for our manuscript. Indeed, the difference of COF mainly comes from the mechanical trapping mechanism of polymer brush, and this is related to the change of interface contact state. As a response, we have added corresponding discussion in the revised manuscript and supplementary information. The real deformation/displacement is the measured displacement minus the reference displacement. And the mechanical trapping evidence is in Response 4.

Comment 3: *The authors compared the shrinkage rates of MALH and NIPAAm before and after phase separation. The author thinks that MALH system can well maintain the material geometry because of the slight volume shrinkage. However, the dramatic shrinkage of the volume will lead to an increase in the density of the polymer brushes*

on the surface, thus improving the lubrication performance. The author should give some explanation.

Response 3: Thank you for your valuable comment. The phase separation mechanisms for MALH and PNIPAAm hydrogels are different. The phase separation of MALH is based on novel strategy of enhancing the electrostatic interaction of the nearby charged residues at high temperature (Fig. 2a), while phase separation of traditional PNIPAAm hydrogels is attributed to the evolution of network water content from switchable hydrogen bonding interaction between polymer chains and water molecules. In this work, our MALH system can well maintain the original material geometry upon encountering phase separation, so there is no dramatic increase of polymer brushes chains density, and the effect of volume effect on lubrication can be ignored. For traditional PNIPAAm hydrogels system, temperature-dominated phase separation would induce large volume change, the dramatic shrinkage of the volume may lead to an increase in the density of the polymer brushes on the surface. However, the mechanical properties of pure PNIPAAm hydrogels system before and after phase separation are poor. If the PNIPAAm hydrogels matrix is used as an artificial lubricating skin for a soft robot, the large deformation will cause geometry mismatch. And the strain energy or stress generated from the deformation may damage the internal components.

Comment 4: *Necessarily, although it is difficult, the authors should provide the modulus data of the surface lubricating layer by micromechanics test.*

Response 4: Thanks for your valuable suggestion. As a response, we have tested the mechanical parameters of the surface lubricating layer in wet state by NHT³ nanoindenter (Probe: Ruby ball; diameter: 1 mm). The surface modulus of the lubricating layer at room temperature was about 0.06MPa, and then increased a little due to the substrate effect and the mechanical trapping effect after phase separation (**Figure R6 a, b**). When the testing loads increase, the obtained modulus is more easily affected by the substrate effect and the mechanical trapping effect (**Figure R6 c, d**). The increase of bulk modulus of MALH caused by phase separation will affect the modulus of the surface lubrication layer. And this result was used for fitting in the Hertz contact model for finite element analysis.

Figure R6 [Supplementary Figure 9]. (a, b) Force-position curves of nanoindentation tests for MALH sample before and after phase separation, and corresponding moduli results (load =50 µN), sample size n=10. (c, d) Force-position curves of nanoindentation tests for control sample without grafting polymer brush and MALH sample before and after phase separation, and corresponding moduli results (load =500 µN), sample size n=10.

Comment 5: *This work compared MALH with other reported modulus-switching systems in supplementary section. The author should give more prominence to the characteristics of thermal stiffening.*

Response 5: we appreciate the reviewer's valuable suggestion. As a response, we have given more prominence to the characteristics of thermal stiffening of MALH in the revised manuscript.

In manuscript: ...Correspondingly, the modulus regulation range of MALH system can match the mechanical properties of diverse biological tissues well, compared with other reported modulus-switching systems (Supplementary Fig. 4, Supplementary Discussion). The modulus-switching mechanism of most traditional materials is based on the crystallization-melting transition or the glass transition. They tend to soften rather than stiffen upon heating, which is opposite to MALH, and the stimulus induced stiffening mechanism in fish skin or other stress reaction in biological system. Furthermore, most of soft robots are usually in soft state at room temperature and stiffen when needed. Therefore, MALH exhibited extraordinary potential application in soft

robots.

Comment 6: *MALH requires external thermal stimulus to achieve regulation. Especially, the authors used hot water as a stimulus. Can other stimuli or remote control be used to achieve regulation? Maybe this can broaden the range of application.*

Response 6: Thank you for such valuable suggestion. In current work, we only used hot water as an external stimulus and didn't consider remote-control route. However, we think realizing the remote-control of phase separation of MALH is completely feasible. Correspondingly, we performed some simple verification experiment. e.g. Some nanoparticles with extraordinary photo-thermal effect, such as Fe_3O_4 , can be added into this thermal-stiffen MALH matrix. As a result, the phase separation behavior of MALH sample can be controlled remotely by near infrared (NIR) laser irradiation. As a proof, we can use NIR laser to write the word "H" on hydrogel. (**Figure R7**).

Figure R7. The phase separation of the Fe_3O_4 nanoparticles-integrated MALH hydrogel controlled by near infrared (NIR) laser irradiation.

Comment 7: *The comparison between MALSD and Ref.50 (Science Robotics, 2021, 6: eabe4088.) should be described in more detail. The fish-like robot in ref.50 improved the swimming efficiency by using a stiffness-tunable tail to control the flow field. In this work, the author needs to highlight the characteristics of the dynamic contact state evolution of the MALSD lead by the switchable modulus.*

Response 7: we appreciate the reviewer's valuable suggestion. As a response, we have described the difference between Ref.50 and MALSD in more detail.

In manuscript: ...At present, most of the bionic fish robots can realize the on-demand control of their soft and rigid moving states by manipulating the mobile joints, so as to change the flow field to achieve efficient drag-reduction.⁵⁰ Different from these bionic fish robots relying on change of mechanical movement states, MALSD can achieve drag reduction through a collaborative strategy of coupling adaptive modulus with surface lubrication. And this strategy is based on mimicking the dynamic contact state between fish and objects. These demonstrations mimicked the escape situation and lubrication enhancement mechanism of fish, and indicated that the MALSD patch can

be used as an intelligent switchable strategy for controlling the movement of mobile devices.

REVIEWERS' COMMENTS

Reviewer #1 (Remarks to the Author):

The authors have satisfactorily addressed my comments.

Reviewer #2 (Remarks to the Author):

The authors addressed all my questions in this revision. I suggest acceptance of this manuscript for publication.

Reviewer #3 (Remarks to the Author):

The authors have well addressed the comments satisfactorily with sufficient experimental support and revision. This manuscript is in a good shape and commensurate with the high level of Nature Communications. It is thus recommended for publication.